# Clinical Outcomes of Patients with Chronic Neuropathic Form of Gaucher Disease in the Spanish Real-World Setting: A Retrospective Study

**DOI:** 10.3390/biomedicines11102861

**Published:** 2023-10-22

**Authors:** Sinziana Stanescu, Patricia Correcher Medina, Francisco J. del Castillo, Olga Alonso Luengo, Luis Maria Arto Millan, Amaya Belanger Quintana, Maria Camprodon Gomez, Lydia Diez Langhetée, Oscar Garcia Campos, Ana Matas Garcia, Jimena Perez-Moreno, Barbara Rubio Gribble, Nuria Visa-Reñé, Pilar Giraldo-Castellano, Mar O’Callaghan Gordo

**Affiliations:** 1Pediatric Metabolic Unit, Hospital Universitario Ramón y Cajal, European Reference Center (MetabERN), 28034 Madrid, Spain; 2Pediatric Nutrition and Metabolic Unit, Hospital Universitario y Politécnico La Fe, 46026 Valencia, Spain; correcher_pat@gva.es; 3Genetics Department, Hospital Universitario Ramón y Cajal, IRYCIS, 28034 Madrid, Spain; fcastillo@salud.madrid.org; 4Centro de Investigación Biomédica en Red de Enfermedades Raras (CIBERER), 28029 Madrid, Spain; 5Pedriatic Unit, Hospital Universitario Virgen del Rocío, 41013 Seville, Spain; 6Internal Medicine Unit, Complejo Asistencial Universitario de León, 24008 León, Spain; 7Rare and Metabolic Diseases Unit, Hospital Universitario Vall d’Hebron, 08035 Barcelona, Spain; camprodon.90@gmail.com; 8Minority Diseases Unit, Hospital de Manises, 46940 Valencia, Spain; 9Pediatric Neurology Unit, Hospital Universitario de Toledo, 45007 Toledo, Spain; 10Muscle Research Unit, Internal Medicine Service, Hospital Clínic de Barcelona, 08036 Barcelona, Spain; 11Pedriatic Unit, Hospital General Universitario Gregorio Marañón, 28007 Madrid, Spain; jimenapermor@gmail.com; 12Pedriatic Unit, Hospital Universitario de Getafe, 28905 Madrid, Spain; 13Pedriatic Unit, Hospital Universitari Arnau de Vilanova, 25198 Lleida, Spain; nvrene.lleida.ics@gencat.cat; 14Fundación Española Para el Estudio y Terapéutica de la Enfermedad de Gaucher y Otras Lisosomales (FEETEG), 50006 Zaragoza, Spain; 15Neurology Unit and Congenital Metabolic Diseases Unit, Hospital Sant Joan de Déu, 08950 Barcelona, Spain; mariamar.ocallaghan@sjd.es

**Keywords:** Gaucher disease type 3, treatment, mutations, clinical manifestations

## Abstract

This was a retrospective, multicenter study that aimed to report the characteristics of type 3 Gaucher disease (GD3) patients in Spain, including the genotype, phenotype, therapeutic options, and treatment responses. A total of 19 patients with GD3 from 10 Spanish hospitals were enrolled in the study (14 men, 5 women). The median age at disease onset and diagnosis was 1 and 1.2 years, respectively, and the mean age at follow-up completion was 12.37 years (range: 1–25 years). Most patients exhibited splenomegaly (18/19) and hepatomegaly (17/19) at the time of diagnosis. The most frequent neurological abnormalities at onset were psychomotor retardation (14/19) and extrinsic muscle disorders (11/19), including oculomotor apraxia, supranuclear palsy, and strabismus. The L444P (c.1448T>C) allele was predominant, with the L444P (c.1448T>C) homozygous genotype mainly associated with visceral manifestations like hepatosplenomegaly, anemia, and thrombocytopenia. All patients received enzyme replacement therapy (ERT); other treatments included miglustat and the chaperone (ambroxol). Visceral manifestations, including hepatosplenomegaly and hematological and bone manifestations, were mostly controlled with ERT, except for kyphosis. The data from this study may help to increase the evidence base on this rare disease and contribute to improving the clinical management of GD3 patients.

## 1. Introduction

Gaucher disease (GD) is a rare disease resulting from a deficiency in the lysosomal enzyme glucocerebrosidase (GCase) due to bi-allelic mutations in the *GBA1* gene [1,2,3]. This deficiency leads to the accumulation of glucocerebroside lipids in one or more organs, resulting in different symptoms [4]. Complex mechanisms such as inflammation, regional cell interaction, and GCase substrate preferences have also been proposed to explain the pathobiology of specific organ involvement and potentially the therapeutic efficacy [1]. The pathophysiological mechanisms of neurological involvement are still not well explained; the process of substrate accumulation in neurons is apparently low, and there is only significant accumulation when the residual GCase activity is extremely low, which happens in some types of *GBA1* mutations [5].

GD is the most common lysosomal storage disease and is classified into three types, depending on the presence and rate of progression of neurologic manifestations [2,3]. The three types are genotypically and phenotypically diverse and differential diagnosis is challenging [6,7]. The most prevalent form of GD is type 1 (OMIM #230800), the non-neuronopathic form. The clinical manifestations are mainly in the hematopoietic system, skeletal system, and visceral organs. The other two types (2 and 3) have early involvement of the central nervous system, with different rates of neurological deterioration. Patients with GD type 2 (OMIM #230900) present with an acute neuronopathic form, with symptoms manifesting either prenatally or during infancy, ultimately resulting in death within the first two years of life. In contrast, GD type 3 (GD3, OMIM #231000) follows a more protracted course and is considered the subacute/chronic neuronopathic form. The incidence of GD3 in the general population is estimated to be less than 1 in 100,000 [8]. Based on the clinical manifestations, GD3 may be subdivided into three subtypes, characterized by predominant neurological involvement (GD3a), internal organ and skeletal involvement (GD3b), and aortic and cardiac valve calcifications (GD3c; cardiovascular type) [9,10]. However, the clinical manifestations often overlap, and this classification is underused in clinical practice. The phenotypic–genotypic heterogeneity in the three types of Gaucher disease is well known; the only indicator that separates neurological from non-neurological forms is the presence of the p.Asn409Ser allele (formerly known as N370S), which excludes the diagnosis of type 2 or type 3 [11].

Comprehensive patient management protocols would be useful to delineate the full phenotypic spectrum of this rare disease, to discern the true characteristics of GD3 patients, and to identify the best clinical and treatment options, to ultimately provide the best possible care to patients with this rare condition. In this regard, different dosages of enzyme replacement and chaperone treatments have been tested, but a consensus on the pharmacological treatment pathways is missing. An international registry of GD patients serves as a reference regarding patient characteristics, treatments, and outcomes [12]. Patients with GD often experience delays in diagnosis, frequent hospitalization and regular follow-up testing, complications of GD, and co-morbidities. Many GD patients all over the world have limited access to medical care and treatment due to their location or inability to pay [13,14]. However, given the particularity of GD3 and the differences in patient management protocols between different healthcare systems, country-specific studies on this population are required. This study aimed to report the characteristics of GD3 patients in Spain, including the genotype, phenotype, therapeutic options, and treatment responses. 

## 2. Materials and Methods

### 2.1. Study Design and Population

This was a retrospective, multicenter study including all patients diagnosed with GD3, followed up at 10 participating centers in Spain. All patients were included in the study after obtaining written informed consent. The patients were diagnosed, treated, and followed according to standard clinical practice in Spain [15]. Patients’ data were collected from the patients’ medical records in an electronic data collection sheet in a pseudonymized fashion between May 2021 and May 2022. Patients were retrospectively followed up from their first medical record since the appearance of the first symptoms (i.e., disease onset) and diagnosis until data collection, defined as the end of the follow-up period. All patients or adult representatives of patients aged <18 years gave written informed consent before any data were collected. This study was conducted according to the local legislation regulations for biomedical research (Law 14/2007), the Helsinki Declaration, and the local Personal Data Protection Law (LOPD 15/1999). The Ethics Committee (CEIC) of the Hospital Sant Joan de Déu (Barcelona, Spain) approved the study protocol (registration number EOM 08-22).

### 2.2. Variables

The primary objective of this study was to describe the clinical outcome and treatment characteristics of patients with GD3.

All data were collected at diagnosis and during follow-up. At baseline, clinical and demographic variables included age, sex, normal pregnancy, and perinatal period (defined as no unexpected event, treatment, or need for hospitalization); family history, including dementias, familial Parkinson’s, and consanguinity, was also considered. Age at disease onset (i.e., the appearance of first symptoms) and at diagnosis were also collected.

Organ manifestations considered variables obtained through clinical examination (hepatomegaly, splenomegaly, characteristic facies, anthropometry, kyphoscoliosis, keeled thorax) or additional tests (hematological, hepatic and bone involvement, cardiopathies, and pulmonary disease). Bone involvement was determined by the presence of bone pain or abnormal radiological findings (X-ray or magnetic resonance (MRI)). Hepatic dysfunction was defined by elevated hepatic transaminase as increased AST/ALT > 1.5 normal values. Hematological disease considered the presence of thrombocytopenia (platelets < 120,000 units/µL) and anemia (hemoglobin (Hb) < 11.5 g/dL). 

The neurological phenotype was determined by the age at onset and neurological manifestations at diagnosis and during follow-up: psychomotor delay, epilepsy, pyramidal (spasticity, increased deep tendon reflexes) and extrapyramidal assessment, ataxia, tremor extraocular movements (slow saccades), and other cranial nerve symptoms (dysphagia, stridor, altered brain stem evoked potentials). A complete ophthalmological examination was performed by an ophthalmologist. 

Treatment variables considered were specific GD medications, including enzyme replacement therapy (ERT), substrate reduction therapy, or chaperone (ambroxol). In addition, other treatments considered were neurologic symptomatic therapies and procedures, including gastrostomy and tracheostomy, walking splints, or a wheelchair.

### 2.3. Statistical Analysis

This study included all patients diagnosed with chronic neuronopathic GD3 in Spain with active follow-up at the beginning of the study, and no sample size calculation was applicable. We performed descriptive analyses. Quantitative variables were presented as the mean and standard deviation (SD), whereas categorical variables were presented as percentages. Statistical analyses were performed using the SAS statistical software, version 9.3 (SAS Institute Inc., Cari, NC, USA).

## 3. Results

### 3.1. Demographic Data

Appendix A summarizes the demographic data of this patient series. A total of 19 patients with GD3 were enrolled in the study, 14 men and 5 women. The mean age of the patients at the end of the follow-up was 12.37 years (range 1–25 years). The median age at disease onset was 1 year (min 0.5, max 20 years), with a median age at diagnosis of 1.2 years. Disease onset and diagnosis occurred during the early years of life (<3 years) in all patients except one, aged 20 years at diagnosis.

Regarding pregnancy history, a significant proportion of patients had a normal pregnancy (14/19 patients; Appendix A); an abnormal course of pregnancy was determined by the abortion risk (one patient), maternal bleeding (one patient) and maternal perinatal fever (two patients). Four patients had a family history of consanguinity and three had other relatives affected by GD3 (one affected brother, three affected cousins). Regarding the familiar background, one patient had a history of Parkinson’s disease and no dementia was recorded. No significant perinatal history was observed, except in one patient who was admitted with vomiting and hypotonia. 

### 3.2. Organ Manifestations

Table 1 outlines the various organ manifestations observed in this series of patients. Most patients exhibited splenomegaly (HP:0001744) (18/19) and hepatomegaly (HP:0002240) (17/19) at the time of diagnosis, while 8/19 patients presented hepatic dysfunction (HP:0002910). Splenectomy was not performed in any of the present patients. Hematological disease was detected at diagnosis, with 9/19 patients presenting thrombocytopenia (HP:0001873) and 11/19 anemia (HP:0001903). The hematological alteration and the hepatic function normalized during the follow-up. Regarding bone involvement, no patient presented bone crisis (HP:0001903). Globally, 5/19 patients (26%) presented bone alteration detected by X-ray or MRI. Bone X-ray was mainly performed at diagnosis (12/19) and detected Erlenmeyer flask bone deformity in two patients. However, MRI was performed at diagnosis in 7/19 patients and detected abnormalities in three cases (bony lytic lesions in one patient and diffuse signal increase at intramedullary level in all long bones in two). During the follow-up, bone MRI was performed in 16/19 patients and detected femoral abnormalities such as slight distal widening in four patients. Bone densitometry was performed in 11/19 patients at the end of the follow-up and detected osteopenia in two cases (z-score lower than −2). At diagnosis, 5/19 (26%) and 3/19 (16%) of patients presented with pectus carinatum (HP:0000768) and kyphoscoliosis (HP:0002751), respectively. 

Severe cardiac involvement consisting of aortic (HP:0004380) and mitral valve calcifications (HP:0004382) was detected in two patients. During the follow-up, an abnormal myocardium morphology (HP:0001637) was found in two more cases, with mild forms of dilated cardiomyopathy in one patient and hypertrophic left ventricle in another. Pulmonary disease was presented in six patients at the end of the follow-up and was mainly conditioned by interstitial involvement (HP:0006530) and restricted pulmonary capacity determined by severe kyphoscoliosis. 

At diagnosis, GD3 patients presented physical features such as short stature (height z-score < −2) (HP:0004322) in 5/19 and microcephaly (head circumference z-score < −2) (HP:0000252) in 2/19. Additionally, at the end of follow-up, 6/19 had characteristic facies, including retro micrognathia (HP:0000308) and low-set pinnae (HP:0000369). 

### 3.3. Neurological Manifestations

Table 2 displays the neurological manifestations observed in this series of GD3 patients. The most common neurological manifestations were abnormalities of horizontal ocular movement, detected in 11/19 at diagnosis and in 14/19 patients at the end of the follow-up; 5/19 patients presented a convergent squint (HP:0000486) and 9/19 ophthalmoplegia (HP:0000602) at diagnosis. Significantly, a psychomotor delay (HP:0001263) was present in 68% (13/19) of the patients at diagnosis, with progression to severe forms during the follow-up. Pyramidal signs such as spasticity and exalted deep tendon reflexes were detected in 5/19 of patients. Extrapyramidal features (parkinsonism, HP:0001300) were detected mainly at the end of the follow-up in three cases. Three of 19 patients presented gait ataxia (HP:0002066), while 5/19 had a tremor (HP:0001337). 

Regarding epilepsy (HP:0001250), another common neurological manifestation, 4/19 (21%) of patients were affected at the time of diagnosis and 6/19 (32%) during the follow-up period. Conversely, myoclonus (HP:0001336) was only observed at the end of the study in 5/19 of patients. 

Cranial nerve manifestations other than abnormal extraocular movement were also significant and were mainly detected at the end of the follow-up, indicating disease progression; 3/19 presented dysphagia (HP:0002015) and 5/19 presented abnormal brainstem auditory evoked potentials (HP:0006958); one patient had severe stridor (HP:0010307).

Hydrocephalus (HP:0000238) and aggressiveness were additional complications that were documented. Lastly, three deaths were recorded in patients with early onset of symptoms in the first year of life and severe neurological progression; two siblings with the [c.1342G>C]; [c.1263del + RecTL] genotype developed parkinsonism during the follow-up and experienced sudden death during sleep; another patient with the [c.1448T>C]; [c.1505G>A] genotype died at the age of 4 years from severe laryngomalacia and apnea crisis. 

### 3.4. Genotypes

Two genetic variants (c.1448T>C and c.1342G>C) accounted for over 70% of the pathogenic alleles (27/38) and appeared both in the homozygous and the compound heterozygous states with one of the remaining nine variants (Table 3 and Table 4). All the pathogenic alleles observed in our cohort corresponded to missense variants, with only one complete loss-of-function variant, c.1263_1317del55 (previously described as del55bp), appearing twice. 

### 3.5. Treatment

All patients received enzyme replacement therapy (ERT) treatment at diagnosis and during the study. The dosing regimens of ERT varied between 60 and 120 UI/kg/2 weeks. Miglustat was administered to a maximum of 11/19 patients during the follow-up, but it was suspended in four of them due to disease progression and/or intolerance. Nine of 19 patients were receiving the chaperone (ambroxol) at the end of the study. Many patients who had been on miglustat were switched to high doses of ambroxol, and this overlapping of treatments made it difficult to determine the efficacy.

The hematological alterations, hepatic dysfunction, and organomegaly were the clinical manifestations that best responded to ERT, as suggested by the significant improvement when starting the treatment after the diagnosis was confirmed. However, neurological manifestations as well as kyphoscoliosis and valvular calcifications did not seem affected and continued their progression during ERT. The miglustat and ambroxol response did not seem to have an evident clinical correlation and was more difficult to assess.

Symptomatic medication (antiepileptic drugs, cardiovascular disease) was used. One patient required gastrostomy and tracheostomy during the follow-up.

## 4. Discussion

This retrospective study collected data from all individuals diagnosed with GD3 with active follow-up in Spain and summarized the characteristics of this population, including demographic data, genotype and phenotype profiles, therapeutic interventions, and the evolution of disease-related organ and neurological manifestations.

Overall, most patients were diagnosed with GD3 during the early years of life, with a representation of both sexes, although male patients were more frequently affected. GD3 was diagnosed early after onset, likely due to the prompt access of patients to the Spanish public healthcare system. However, other studies have reported the delayed diagnosis of GD patients due to different factors [27,28]. Among these factors, the most relevant are the rarity of the disease and its presentation, with non-specific symptoms overlapping with those of more common diseases in childhood, such as hematologic malignancy and infectious diseases [6]. Additionally, disseminating knowledge about GD3 to other healthcare professionals is crucial, given that neuropediatricians, ophthalmologists, orthopedic/traumatologists, and hematologists may be the first to care for these patients.

The clinical heterogeneity of GD3, with no specific clinical signs/symptoms and wide clinical manifestation variation, poses challenges in recognizing the disease, especially in its later forms, and distinguishing between different phenotypes [28]. Notably, discriminating between GD1 and late GD3 or between late GD2 and early GD3 is not straightforward [2,29,30]. To address these challenges, diagnostic algorithms may improve the early diagnosis of patients with GD3, enabling the initiation of targeted treatments as early as possible [7,31,32]. Currently, there are several diagnostic algorithms available for GD that are based on the assessment of clinical symptoms, laboratory analysis, and genetic evaluation [31]. These algorithms have proven useful in identifying patients with GD, and some studies have suggested that they may improve the rate of diagnosis [7,33]. However, it is important to keep in mind that GD is a rare and complex disease, and early and accurate identification may be difficult given the variability of symptoms and lack of disease awareness among physicians and other healthcare professionals. Therefore, while diagnostic algorithms can be helpful in increasing the rate of diagnosis or shortening the delay in diagnosis, it is also important for physicians to have a thorough understanding of the disease and to be alert to possible symptoms in at-risk patients. To this end, monitoring neurological signs and symptoms during childhood is imperative for all types of GD. Neurodiagnostic techniques, such as computed tomography (CT), magnetic resonance imaging (MRI), and electrophysiology, reveal alterations that are secondary to advanced neurological damage but are not useful in making a differential diagnosis at the early disease stages [34]. In this regard, a genetic diagnosis is helpful, as some genotypes are more commonly associated with a particular clinical form [8]. However, patients with the same genotype, including siblings and identical twins, such as some of the patients in this series, may have different clinical presentations and outcomes due to various factors [35,36].

Visceral manifestations were mostly detected at diagnosis and were controlled with ERT, particularly organomegaly and hematological and hepatic manifestations. However, other peripheral manifestations, such as those described within the D409H (c.1342G>C) genotype in homozygosis (cardiovascular calcifications or corneal opacities), appeared during follow-up. These manifestations progressed and had an unclear relationship with the treatment administered, as previously reported by Kurolap et al. [37]. Bone involvement was mild in GD3 patients, unlike the GD1 phenotype, which classically presents severe pain crisis at diagnosis. Regarding kyphosis, it had severe progression in our study and apparently was not related to the ERT, leading to impaired pulmonary capacity secondary to the important restriction; thus, many authors consider kyphosis one of the neurological manifestations in GD3 patients due to its severity and poor evolution [34] and one of the signs that can help to diagnose GD3 [38,39]. Moreover, this manifestation is associated with the Norrbottnian clinical variant of GD [33,39]. Growth retardation and microcephaly were also observed. In some cases, growth retardation may persist even after several years of treatment [30]. Long-term studies are needed to evaluate the long-term efficacy and safety of bone therapy in patients with GD3. In summary, although some of the various organ manifestations of GD3 may be controlled with ERT, others may be progressive and severe.

In this patient series, the most frequent neurological abnormalities were psychomotor retardation (14/19) and extrinsic muscle disorders (11/19), including oculomotor apraxia, supranuclear palsy, and strabismus. As expected, psychomotor retardation symptoms worsened throughout the follow-up period. The involvement of extraocular muscles (EOM) was the most frequent neurological sign at diagnosis, in line with previous studies [12]. Although neurosensorial deafness, defined as abnormal BAEP, is not a common manifestation in the neuropathic forms of GD, it was frequent in our series. Sensorineural hearing loss in GD3 is attributed to a build-up of glucocerebroside in the nerve cells of the inner ear and in the auditory nerves that transmit auditory impulses to the brain [40,41]. Importantly, sensorineural deafness in GD3 may be progressive and, therefore, regular audiological assessments are essential to monitor hearing [41].

However, in some cases, neurological abnormalities may appear later in the disease course, after hematologic and visceral manifestations [30]. In our series, patients with neurological symptoms at baseline had more severe involvement and an earlier diagnosis, highlighting the wide clinical variability of GD3 manifestations. Regarding the progression of symptoms, while hematological and visceral manifestations are common at the time of diagnosis, neurological abnormalities may appear within the first 1–2 years in most registries [12,42,43]. In summary, the classic phenotype of GD3 is characterized by a combination of neurological and visceral symptoms, the former being the most evident [31]. Patients may have wide variability in the severity and progression of symptoms, and the onset of symptoms may be in childhood, adolescence, or adulthood. Neurological variability in GD3 is an important aspect as it can influence the course of the disease in individual patients, their prognosis, and their response to treatment [44].

Regarding mortality, three cases in our series corresponded to patients with a severe neurological disease who presented epilepsy and moderate psychomotor delay at diagnosis and who developed parkinsonism, myoclonus, and bulbar involvement early on. Two of these cases responded well to treatment with L-Dopa, with improved extrapyramidal symptoms, as previously reported by Darling and collaborators [45].

Not surprisingly, two well-known variants associated with neurological symptoms in GD (c.1448T>C and c.1342G>C) constituted more than 70% of the disease-causing alleles (27/38). Of note, all the observed disease-causing alleles in our study were missense variants, except for one complete loss-of-function variant (c.1263_1317del55), which appeared in two siblings with a severe form [45]. This may be in accordance with one of the molecular models for neuronopathic GD, which posits that neurological symptoms arise from likely gain-of-function mechanisms caused by amino acid substitutions at critical glucocerebrosidase residues [46]. Previous studies have linked the [c.1448T>C]; [c.1448T>C] homozygous genotype to severe neurological symptoms in Japanese and Egyptian patients [30,47,48]. However, in our series of patients, this genotype, observed in 7/19 patients, was primarily associated with visceral manifestations such as hepato-splenomegaly and anemia/thrombocytopenia, with minimal neurological or ocular muscle involvement, with a median follow-up of 8 years. This raises the question of whether disease modification could be achieved through treatments during this time lag between visceral and neurological symptoms. The phenotype of the two [c.1342G>C]; [c.1342G>C] homozygotes is characterized by corneal opacities, hydrocephalus, valvular calcifications (aortic/mitral), and hepatosplenomegaly, but minimal or no primary hematologic or neurologic manifestations, and was first described in a Spanish family [49]. The prognosis of these patients seems determined by the cardiovascular involvement, which required valvular replacement in both patients. In contrast, c.1342G>C compound heterozygotes lack valvular calcifications or corneal opacities, with a phenotype likely determined by the clinical impact of the other allele [50]. Hence, we observed very severe neurologic involvement in the two [c.1483G>C] + [c.1497G>C] + [c.1263_1317del55] + [c.1342G>C] compound heterozygotes, which led to early deaths. Conversely, the only [c.1448T>C]; [c.1342G>C] compound heterozygote had a milder neurological phenotype and more hematologic and skeletal involvement. Interestingly, the genotype [c.475C>T]; [c.680A>G] was associated with a peculiar phenotype of organomegaly and persistent absent seizure, with good control with chaperone (ambroxol) treatment. Despite efforts to unify the criteria for the clinical and prognostic evaluation of patients with neuropathic forms of GD, phenotype–genotype correlation is still considerably heterogeneous [1]. Indeed, the better genetic characterization of patients, including at least a complete analysis of the *GBA1*-*GBAP1* locus at chromosome 1 and screening for variants at genes encoding proteins that modify glucocerebroside activity, such as *PSAP* and *GRN*, should help to establish better correlations in the future in these patients [50,51].

ERT is currently the only approved specific treatment for the pediatric population [50]. All patients in this series received ERT therapy (imiglucerase) at variable doses, ranging from 60 to 120 U/kg/2 weeks. ERT is not able to cross the blood–brain barrier, precluding a direct impact on CNS manifestations [52]. However, it may reduce neuroinflammation by reducing the excessive microglial response to proinflammatory cytokines and chemokines, although confirmatory studies are required [1]. As expected, hematological manifestations, organomegaly, and hepatic dysfunction responded very well to ERT in this series of GD3 patients [15,28,53,54]. However, ERT had no effect on the progression of severe kyphoscoliosis or interstitial pulmonary involvement. Other visceral manifestations that remained unmodified by treatment in our series were the valvular calcifications and corneal opacities detected in c.1342G>C homozygous patients. Moreover, patients with valvular calcifications required valve replacement techniques, and hydrocephalus required ventriculoperitoneal shunt placement.

Patients with GD3 can show a wide degree of variation in the neurological progression and severity of the systemic disease, making the assessment of outcomes with available therapies difficult. Although the results of the clinical trial with miglustat did not show the expected efficacy, we must be aware that neurological damage is irreversible and the objective in these patients is to prevent further damage [55]. Oral ambroxol at high doses had good safety and tolerability, significantly increased lymphocyte glucocerebrosidase activity, permeated the blood–brain barrier, and decreased glucosylsphingosine levels in cerebrospinal fluid, producing a slowing in the progression of neurological symptoms in some chaperone-sensitive mutants [56].

All of the available evidence suggests that there are still therapeutic targets to be achieved in patients with GD3, a finding that underscores the need for continued research in this area.

This study has several limitations, of which the most relevant are associated with the nature of the disease, with low prevalence and high genotypic and phenotypic disparity. However, despite the disease’s low prevalence, we included a total of 19 GD3 patients from different regions of Spain in the study. Another limitation is the use of a retrospective registry, based on medical records that were not originally designed to collect data for research, resulting in some missing information. However, the databases were controlled, and data collection was monitored with the aim of minimizing the impact of this limitation. Finally, the study was not entirely comprehensive as it did not evaluate the patient/caregiver burden and quality of life, an essential aspect highlighted by recent findings [57]. Upcoming research is anticipated to address this omission.

## 5. Conclusions

To our knowledge, this registry of GD3 patients is the most detailed series in terms of the clinical and neurological characteristics in Spain. The study considered various aspects of the disease, such as demographic attributes, genotype and phenotype profiles, therapeutic options, and treatment outcomes. The neurological variability in GD3 is in part due to genetic heterogeneity and the complexity of the molecular mechanisms underlying the disease. Epigenetic and environmental factors may also play a role, making the presentation of the disease unique to each patient. The data from this study may be compared to other GD3 registries, helping to increase the body of evidence on this rare disease and contributing to improving the clinical management of GD3 patients.

## 6. Patents

Does not apply.

## Figures and Tables

**Table 1 biomedicines-11-02861-t001:** Organ manifestations.

		At Diagnosis		At the End of the Follow-Up	
		n	%	n	%
**Hepatomegaly ^1^**		17	89	2	11
**Splenomegaly ^2^**		18	95	2	11
**Hematological**	**Thrombocytopenia ^3^**	9	47	0	0
	**Anemia ^4^**	11	58	0	0
**Hepatic dysfunction ^5^**		8	42	0	0
**Bone crisis**		0	0	0	0
**Kyphoscoliosis**		3	16	3	16
Pectus carinatum		5	26	6	32
**Bone involvement**	**X-ray** **Bone MRI**	2 3	10 15	- 4	- 21
**Cardiopathies**		1 ^6^	5	4 ^7^	21
**Pulmonary disease**		2	11	6	32
**Corneal opacities**		1	5	3	16
**Characteristic facies**		3	16	6	32
**Anthropometry**	**Low height (z-score < −2)**	5	26	5	26
	**Low weight (z-score < −2)**	5	26	3	16
	**Microcephaly (HC z-score < −2)**	2	11	1	5

^1^ Males aged 5–12 years liver >3.5%, and when >12 years, liver >2.5% of body weight; females aged 5–12 years liver >3.2%, and when >12 years, liver >2.9% of body weight. The equivalence is 1 g/mL of hepatic volume. ^2^ Spleen 0.2% of body weight. Equivalence is 0.45–0.6 g/mL. ^3^ PLT < 120,000/µL. ^4^ Hb < 11.5 g/dL. ^5^ AST/ALT > 1.5 NV. ^6^ Calcifications. ^7^ Calcifications (n = 2), mild dilated cardiomyopathy (n = 1), and mild hypertrophic cardiomyopathy (n = 1). HC, head circumference; MRI, magnetic resonance imaging.

**Table 2 biomedicines-11-02861-t002:** Neurological manifestations.

		At Diagnosis		At the End of the Follow-Up	
		n	%	n	%
Epilepsy		4	21	6	32
Myoclonus		0		5	26
Extrapyramidal features (parkinsonism)		0		3	16
Pyramidal signs (spasticity, exalted DTR)		5	26	5	26
Ataxia		1	5	3	16
Tremor		0		5	26
Cranial nerve manifestations ^1^ Dysphagia Altered BAEP Stridor		1 5 0	5 26	4 5 1	21 26 5
Extraocular muscle assessment Convergent squint Ophthalmoplegia (supranuclear palsy, oculomotor apraxia)		5 9	26 47	6 11	31 57
Psychomotor retardation	Mild	7	37	5	26
	Moderate	6	31	5	38
	Severe	0		3	23
March released		5	26	8	42
Normal language		9	47	10	53
Others ^2^				2	10

BAEP, brainstem auditory evoked potential; DTR, deep tendon reflexes. ^1^ Other than extraocular muscles; ^2^ hydrocephalus (n = 1) and aggressiveness (n = 1).

**Table 3 biomedicines-11-02861-t003:** Genotype and phenotype correlation. M: male; F: female; HS: hepatosplenomegaly; †: exitus.

Patient	Gender/Current Age (Years)	Genotype	Hepatosplenomegaly	Hematological	Epilepsy	Extraocular Muscles	Neurological Manifestations	Bone Manifestations	Cardiopathies	Pulmonary
1	M/9	c.754T>A; c.1093G>A; c.1448T>C F214I/E326K/L444P	H1S1	Anemia, thrombocytopenia	Yes	Yes	Psychomotor delay	Kyphoscoliosis Erlenmeyer deformity at diagnosis	Yes (mild left ventricular hypertrophy)	Yes (interstitial and restrictive)
2	M/16	c.1342G>C; c.721G>A D409H/G241R	H1S1	No	No	Yes	No	Kyphoscoliosis	No	No
3	M/25	c.1342G>C; c.1342G>C D409H/D409H	H1S1	No	No	Yes	No	Bone MRI: diffuse signal increase in long bones	Yes (calcifications)	no
4	M/23	c.1342G>C; c.1342G>C D409H/D409H	H1S1	Anemia	No	No	Hydrocephalus	No	Yes (calcifications)	No
5	F/4	c.1448T>C; c.1448T>C L444P/L444P	H1S1	Anemia, thrombocytopenia	No	No	Tremor	No	No	Yes (interstitial)
6	M/10	c.475C>T; c.680A>G R159W/N228S	H1S1	No	Yes	No	No	No	No	No
7	M/22	c.1448T>C; c.1342G>C L444P/D409H	H1S1	Thrombocytopenia	No	Yes	Psychomotor delay, ataxia	No	No	No
8	M/27	c.1505G>A; c.1246G>A R463H/G377S	H1S1	Anemia	No	Yes	Psychomotor delay	No	No	No
9	M/8	c.1448T>C; c.1448T>C L444P/L444P	H1S1	Anemia, thrombocytopenia	No	No	Psychomotor delay	Kyphoscoliosis Erlenmeyer deformity at diagnosis Lytic lesions	No	Yes (restrictive)
10	M/21	c.1448T>C; c.1448T>C L444P/L444P	H1S1	Anemia, thrombocytopenia	Yes	Yes	Psychomotor delay, aggressiveness, ataxia	Bone MRI: diffuse signal increase in long bones	No	No
11	F/11	c.1448T>C; c.1448T>C L444P/L444P	H1S1	Thrombocytopenia	Yes	No	Psychomotor delay	No	No	No
12	M/17	c.1049A>G; c.1193G>A H350R/R359Q	H1S1	Anemia	No	Yes	Psychomotor delay	No	No	No
13	F/10	c.1448T>C; c.1448T>C L444P/L444P	H0S1	Thrombocytopenia	No	Yes	Thinning or corpus callosum on cranial MRI	No	No	No
14	M/6 †	c.1342G>C; c.1263_1317del D409H/del 55pb	H1S1	Thrombocytopenia	No	Yes	Psychomotor delay, parkinsonism	No	No	Yes
15	F/7 †	c.1342G>C; c.1263_1317del D409H/del 55pb	H0S0	No	No	Yes	Psychomotor delay, parkinsonism	No	No	Yes
16	F/2	c.1448T>C; c.1342G>C L444R/D409H	H1S1	Anemia	No	Yes	No	No	No	No
17	M/5	c.1448T>C; c.1448T>C L444P/L444P	H1S1	Anemia, thrombocytopenia	No	No	Delay in expressive language	No	Mild left ventricular hypertrophy	No
18	M/23	c.1448T>C; c.1448T>C L444P/L444P	H1S1	Anemia	Yes	Yes	Psychomotor delay	No	No	No
19	M/4 †	c.1448T>C; c.1505G>A L444P/R463H	H1S1	No	Yes	Yes	Psychomotor delay, dysphagia, stridor	No	No	Yes

**Table 4 biomedicines-11-02861-t004:** Genotypes.

Variant (Current HGVS Rules)	Protein	Reported to the Patient as	# of Alleles	Original Citation
c.1448T>C	p.Leu483Pro	L444P	18	[16]
c.1342G>C	p.Asp448His	D409H	9	[17]
c.1263_1317del	p.Leu422Profs *4	del55pb	2	[18]
c.1505G>A	p.Arg502His	R463H	2	[19]
c.475C>T	p.Arg159Trp	R159W	1	[20]
c.680A>G	p.Asn227Ser	N228S	1	[21]
c.721G>A	p.Gly241Arg	G241R	1	[22]
c.754T>A	p.Phe252Ile	F214I	1	[23]
c.1049A>G	p.His350Arg	H350R	1	[24]
c.1193G>A	p.Arg398Gln	R359Q	1	[25]
c.1246G>A	p.Gly416Ser	G377S	1	[26]

HGVS = Human Genome Variation Society. # Number.

## Data Availability

Complete data of this study are available in the Appendix A.

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
