# Peer review of "Clinical Outcomes of Patients with Chronic Neuropathic Form of Gaucher Disease in the Spanish Real-World Setting: A Retrospective Study"

_biomedicines, 2023, doi:10.3390/biomedicines11102861_

Round 1
Reviewer 1 Report
The l article by Stanescu et al. "Clinical Outcomes of Patients with Chronic Neuropathic Form of Gaucher Disease in the Spanish Real-World Setting: A Retrospective Study" covers a potentially interesting and emerging topic related to the Gaucher Disease therapy. In this sense, this remains to be potentially interesting for the biomedicines readers.
I regard the main point of this paper as highly attractive as well as the results are clearly presented. The text does not contain any major errors, therefore I have some minor comments and recommendations:
1. There is a need to provide slightly more expanded introduction shortly
mentioning/describing pathogenesis of Gaucher disease and its impact of modern healthcare.
2. The figure summarizing and clarifying the results should be added.
3. Following references should be added and properly cited within the main text:
- Mela A, Rdzanek E, Tysarowski A, Sakowicz M, Jaroszyński J, Furtak-Niczyporuk M, Żurek G, Poniatowski ŁA, Jagielska B. The impact of changing the funding model for genetic diagnostics and improved access to personalized medicine in oncology. Expert Rev Pharmacoecon Outcomes Res. 2023 Jan;23(1):43-54. doi: 10.1080/14737167.2023.2140139.
- Huang YN, Huang JY, Liao WL, Chiang SL, Liu KW, Bau DT, Wang CH, Su PH. Incidence of Pulmonary and Respiratory Conditions in Gaucher Disease from 2000 to 2020: A Multi-institutional Cohort Study. In Vivo. 2023 Sep-Oct;37(5):2276-2283. doi: 10.21873/invivo.13330.
- Mela A, Poniatowski ŁA, Drop B, Furtak-Niczyporuk M, Jaroszyński J, Wrona W, Staniszewska A, Dąbrowski J, Czajka A, Jagielska B, Wojciechowska M, Niewada M. Overview and Analysis of the Cost of Drug Programs in Poland: Public Payer Expenditures and Coverage of Cancer and Non-Neoplastic Diseases Related Drug Therapies from 2015-2018 Years. Front Pharmacol. 2020 Aug 14;11:1123. doi: 10.3389/fphar.2020.01123.
- Bettioui T, Chipeaux C, Ben Arfa K, Héron S, Belmatoug N, Franco M, de Person M, Moussa F. Development of a new online SPE-HPLC-MS/MS method for the profiling and quantification of sphingolipids and phospholipids in red blood cells - Application to the study of Gaucher's disease. Anal Chim Acta. 2023 Oct 16;1278:341719. doi: 10.1016/j.aca.2023.341719.
4. In some places the use of English could be improved on.
Completing this gaps will have an impact on the understanding the aim of the study and, frommy point of view, is absolutely necessary.
minor review
Author Response
The l article by Stanescu et al. "Clinical Outcomes of Patients with Chronic Neuropathic Form of Gaucher Disease in the Spanish Real-World Setting: A Retrospective Study" covers a potentially interesting and emerging topic related to the Gaucher Disease therapy. In this sense, this remains to be potentially interesting for the biomedicines readers.
I regard the main point of this paper as highly attractive as well as the results are clearly presented. The text does not contain any major errors, therefore I have some minor comments and recommendations:
- There is a need to provide slightly more expanded introduction shortly
mentioning/describing pathogenesis of Gaucher disease and its impact of modern healthcare.
Thank you for this accurate commentary. According your suggestion we have improve the description of pathogenesis of Gaucher disease in the introduction section with the following paragraphs that we have discussed in discussion section.
Line 53 “Complex mechanism such as inflammation, regional cell interaction or GCase substrate preferences have also been proposed to explain the pathobiology of specific organ involvement and potentially the therapeutic efficacy (1)”.
Line 55 “The pathophysiological mechanisms of neurological involvement are still not well explained; the process of substrate accumulation in neurons is apparently low, and there is only significant accumulation when the residual GCase activity is extremely low, which happens in some types of GBA1 mutations (Ref Orvisky E et al)”.
Line 73 “The phenotypic-genotypic heterogeneity in the three types of Gaucher disease is well known, the only indicator that separates neurological from non-neurological forms is the presence of the p.Asn409Ser allele (formerly known as N370S) excludes the diagnosis of type 2 or type 3” (Schiffmann R et al)”.
- The figure summarizing and clarifying the results should be added.
We have added a more complete description of the sample in a new table (Table 3 in the main text, former Table 3 in supplementary material), according to the suggestion of the other referees. We believe that the results are better summarized and clarified in this format, due to the characteristics of our samples.
- Following references should be added and properly cited within the main text:
- Mela A, Rdzanek E, Tysarowski A, Sakowicz M, Jaroszyński J, Furtak-Niczyporuk M, Żurek G, Poniatowski ŁA, Jagielska B. The impact of changing the funding model for genetic diagnostics and improved access to personalized medicine in oncology. Expert Rev Pharmacoecon Outcomes Res. 2023 Jan;23(1):43-54. doi: 10.1080/14737167.2023.2140139.
- Huang YN, Huang JY, Liao WL, Chiang SL, Liu KW, Bau DT, Wang CH, Su PH. Incidence of Pulmonary and Respiratory Conditions in Gaucher Disease from 2000 to 2020: A Multi-institutional Cohort Study. In Vivo. 2023 Sep-Oct;37(5):2276-2283. doi: 10.21873/invivo.13330.
- Mela A, Poniatowski ŁA, Drop B, Furtak-Niczyporuk M, Jaroszyński J, Wrona W, Staniszewska A, Dąbrowski J, Czajka A, Jagielska B, Wojciechowska M, Niewada M. Overview and Analysis of the Cost of Drug Programs in Poland: Public Payer Expenditures and Coverage of Cancer and Non-Neoplastic Diseases Related Drug Therapies from 2015-2018 Years. Front Pharmacol. 2020 Aug 14;11:1123. doi: 10.3389/fphar.2020.01123.
- Bettioui T, Chipeaux C, Ben Arfa K, Héron S, Belmatoug N, Franco M, de Person M, Moussa F. Development of a new online SPE-HPLC-MS/MS method for the profiling and quantification of sphingolipids and phospholipids in red blood cells - Application to the study of Gaucher's disease. Anal Chim Acta. 2023 Oct 16;1278:341719. doi: 10.1016/j.aca.2023.341719.
We have added the reference Mela A et al (2020) appropriate to the purpose of the manuscript.
- In some places the use of English could be improved on.
The English was corrected by an English medical writer
Completing this gaps will have an impact on the understanding the aim of the study and, frommy point of view, is absolutely necessary.
Reviewer 2 Report
This retrospective study reports genotypes, phenotypes, and treatment responses in patients with type 3 of Gaucher disease. This kind of study has merit, but the present manuscript requires some improvements before it can be accepted for publication.
1. In Table S3, it would be useful for readers if the authors could include genotypes and phenotypes for all patients. In the main text, it should be clearly described whether there is a correlation between genotypes and phenotypes.
2. In section 3.6, there is very limited information on treatment responses. Although Table 2 summarizes neurological manifestations, it is unclear whether there is an improvement after ERT. The responses after miglustat and ambroxol administration are also unclear. These issues should be clearly described and discussed.
3. I have problems with some numbers of patients in Table 2. According to the authors, a total of 19 patients were recruited. However, the sum of patients with extraocular muscle assessment or with psychomotor retardation is greater than 19.
Author Response
This retrospective study reports genotypes, phenotypes, and treatment responses in patients with type 3 of Gaucher disease. This kind of study has merit, but the present manuscript requires some improvements before it can be accepted for publication.
- In Table S3, it would be useful for readers if the authors could include genotypes and phenotypes for all patients. In the main text, it should be clearly described whether there is a correlation between genotypes and phenotypes.
Thank you for this accurate commentary. Really the table 3 complete with the description of genotype/phenotype of all patients is more informative and interesting for the readers. We have included a new table 3 with all cohort. In section 3.6, there is very limited information on treatment responses.
Although Table 2 summarizes neurological manifestations, it is unclear whether there is an improvement after ERT. The responses after miglustat and ambroxol administration are also unclear. These issues should be clearly described and discussed.
All type 3 patients have received treatment with ERT, in 11 patients miglustat was added to the enzyme for compassionate use and although the results have not been totally satisfactory, slowing of the progression of neurological manifestations and in some patients control of epileptic episodes has been observed. The same occurs with the administration of ambroxol, which in some patients produces a containment of neurological manifestations, but it is difficult to evaluate due to the overlapping between the different treatments, plus the adjustment of the neuroleptic drugs.
We have added a paragraph with this description in the results section and an additional comment in the discussion section according to other similar publications.
3. I have problems with some numbers of patients in Table 2. According to the authors, a total of 19 patients were recruited. However, the sum of patients with extraocular muscle assessment or with psychomotor retardation is greater than 19.
Thank you very much for this comment, we have reviewed and corrected these data.
Round 2
Reviewer 2 Report
My comments have been addressed in a satisfactory manner.